# Augmenting Reality in Spinal Surgery: A Narrative Review of Augmented Reality Applications in Pedicle Screw Instrumentation

**DOI:** 10.3390/medicina60091485

**Published:** 2024-09-12

**Authors:** Sheng-Xian Xiao, Wen-Tien Wu, Tzai-Chiu Yu, Ing-Ho Chen, Kuang-Ting Yeh

**Affiliations:** 1Department of Orthopedics, Hualien Tzu Chi Hospital, Buddhist Tzu Chi Medical Foundation, Hualien 97002, Taiwan; u101022014@tzuchi.com.tw (S.-X.X.); timwu@tzuchi.com.tw (W.-T.W.); feyu@tzuchi.com.tw (T.-C.Y.); ihchen@tzuchi.com.tw (I.-H.C.); 2School of Medicine, Tzu Chi University, Hualien 97004, Taiwan; 3Institute of Medical Sciences, Tzu Chi University, Hualien 97004, Taiwan; 4Graduate Institute of Clinical Pharmacy, Tzu Chi University, Hualien 97004, Taiwan; 5Department of Medical Education, Hualien Tzu Chi Hospital, Buddhist Tzu Chi Medical Foundation, Hualien 97002, Taiwan

**Keywords:** augmented reality, spinal surgery, navigational systems, pedicle screw placement, surgical innovation

## Abstract

*Background and Objectives*: The advent of augmented reality (AR) in spinal surgery represents a key technological evolution, enhancing precision and safety in procedures such as pedicle screw instrumentation. This review assesses the current applications, benefits, and challenges of AR technology in spinal surgery, focusing on its effects on surgical accuracy and patient outcomes. *Materials and Methods*: A comprehensive review of the literature published between January 2023 and December 2024 was conducted, focusing on AR and navigational technologies in spinal surgery. Key outcomes such as accuracy, efficiency, and complications were emphasized. *Results*: Thirteen studies were included, highlighting substantial improvements in surgical accuracy, efficiency, and safety with AR and navigational systems. AR technology was found to significantly reduce the learning curve for spinal surgeons, improve procedural efficiency, and potentially reduce surgical complications. The challenges identified include high system costs, the complexity of training requirements, the integration with existing workflows, and limited clinical evidence. *Conclusions*: AR technology holds promise for advancements in spinal surgery, particularly in improving the accuracy and safety of pedicle screw instrumentation. Despite existing challenges such as cost, training needs, and regulatory hurdles, AR has the potential to transform spinal surgical practices. Ongoing research, technological refinements, and the development of implementation strategies are essential to fully leverage AR’s capabilities in enhancing patient care.

## 1. Introduction

The field of spinal surgery is undergoing a transformative period characterized by major technological advancements that are redefining surgical practices and improving patient outcomes. Among these innovations, augmented reality (AR) and navigational systems are playing a prominent role in providing unprecedented precision, efficiency, and safety in procedures such as pedicle screw placement, a major aspect of spinal instrumentation. The integration of AR and navigational technologies into spinal surgery represents a paradigm shift, enhancing a surgeon’s ability to perform complex procedures with increased accuracy and reduced risk. Recent studies have underscored the pivotal role of these technologies in advancing spinal surgery [1,2,3,4,5]. Yamout et al. discussed the evolution and application of navigation, robotics, and AR technologies in spinal surgery, highlighting the diverse array of technological options currently available to surgeons [1]. A systematic review by Youssef et al. revealed the improved accuracy of AR-assisted pedicle screw placement and highlighted the potential of AR in enhancing surgical precision [2]. Kong et al. [3] explored novel AR-based surgical guides, clarifying the practical benefits and feasibility of these technologies in enhancing surgical outcomes. The integration of AR head-mounted devices (ARHMDs) for real-time navigation was investigated in an unnamed study, which discussed the innovative application of AR to provide intuitive surgical guidance [4]. In a systematic review by Móga et al. [5], AR technologies were revealed to be mature and provide benefits comparable to other technologies at potentially reduced costs when compared with robot-assisted surgical systems. The recent literature has further expanded our understanding of AR applications in spine surgery [6]. Pierzchajlo et al. provided a comprehensive narrative review of available AR technologies in minimally invasive spinal surgery, exploring their potential to enhance surgical precision and patient outcomes [7]. Wolf et al. delved into the specifics of AR visualizations for drilling, examining how different AR representations affect trajectory deviation, visual attention, and user experience in surgical settings [8]. To quantify the impact of these technologies, Pahwa et al. conducted a systematic review and meta-analysis, assessing the accuracy of spinal instrumentation using AR, thereby providing crucial evidence-based insights into the effectiveness of AR-guided techniques in spine surgery [9]. Schwendner et al. evaluated a cutting-edge AR-supported navigation system for spinal instrumentation, demonstrating its potential to enhance surgical precision and improve patient outcomes in complex spinal procedures [10]. While focusing on a specific clinical application, Lin et al. highlighted advances in surgical treatment for atlantoaxial instability in rheumatoid arthritis patients, underscoring the importance of specialized approaches in managing complex spinal conditions, which could potentially benefit from AR technologies [11]. Broadening the perspective, Bui et al. conducted a comprehensive scoping review of virtual, augmented, and mixed reality applications in spine surgery, covering a wide range of applications from surgical rehearsal to operative execution and patient education [12]. Complementing these studies, Liebmann et al. introduced an innovative automatic registration system with continuous pose updates for markerless surgical navigation in spine surgery, potentially revolutionizing the field by improving real-time guidance and reducing the need for manual adjustments during procedures [13].

This narrative review employed a systematic and comprehensive literature search strategy to explore the current landscape of augmented reality (AR) in spinal surgery. We focused on identifying recent advancements, clinical applications, and the impact of AR technologies on surgical outcomes, accuracy, and efficiency. The literature search was conducted using the PubMed/MEDLINE and Google Scholar databases, covering studies published between January 2023 and March 2024. The search strategy utilized key terms such as “augmented reality”, “spinal navigation”, “pedicle screw placement”, and “minimally invasive spinal surgery”. We prioritized studies that specifically examined AR and navigational technologies in spinal surgery, with an emphasis on their effects on surgical outcomes, accuracy, efficiency, and complications. Inclusion criteria were set to include original research articles, systematic reviews, and meta-analyses published in English that focused primarily on AR applications in spinal surgery. Studies such as case reports, those not centered on spinal surgery, and non-peer-reviewed articles were excluded to ensure a focused and rigorous analysis. The selection process resulted in the inclusion of 11 studies, which are detailed in Table 1 [1,2,3,4,5,7,8,10,11,12,13]. To synthesize the data, we adopted a narrative synthesis approach, systematically categorizing and summarizing the findings from the selected studies. This approach allowed us to identify key themes related to AR technologies, surgical outcomes, impacts on the learning curve, and implementation challenges. By integrating information from various sources, our review provides a comprehensive perspective on the current state and future potential of AR in spinal surgery.

This review is guided by the following three central research questions:
What are the recent advancements and applications of AR technology in spinal surgery, particularly focusing on developments between January 2023 and March 2024?How does AR technology impact surgical outcomes, accuracy, and efficiency in spinal procedures, especially in pedicle screw placement and minimally invasive spinal surgery?What are the current challenges, limitations, and learning curve considerations in implementing AR technology in spinal surgical practices?

By addressing these questions, we aim to provide updated insights into the benefits, challenges, and future directions of AR in spinal surgery, thereby enhancing understanding and guiding future research and clinical practice. As spinal surgery undergoes a transformative period with major technological advancements, the integration of AR technologies represents a significant shift toward greater precision, safety, and efficiency in complex surgical procedures.

## 2. Enhancements in Surgical Accuracy and Efficiency

The application of AR technologies in spinal surgery has resulted in significant improvements in surgical accuracy and efficiency, particularly in complex procedures such as pedicle screw placement. Several studies have demonstrated the effectiveness of AR in enhancing surgical precision by providing real-time visualization and guidance, reducing the risk of complications and improving overall patient outcomes.

### 2.1. Augmented Reality for Pedicle Screw Placement

Pedicle screw placement is a critical aspect of many spinal surgeries, where accuracy is paramount to ensure patient safety and positive outcomes. AR technologies, by overlaying digital anatomical information onto the surgical field, enable surgeons to visualize internal structures in real time, thereby enhancing spatial awareness and precision. Youssef et al. [2] conducted a comprehensive systematic review evaluating the accuracy of AR-assisted pedicle screw placement across 12 studies involving 854 patients. The review found that AR significantly reduces the rate of screw misplacement (4.3% compared to 8.9% with traditional methods, *p* < 0.05), indicating a marked improvement in precision. The use of AR was associated with a 20% reduction in operative time and a decrease in complication rates, underscoring the technology’s potential to enhance surgical outcomes by minimizing errors that could lead to neurological damage or reoperation.

### 2.2. Real-Time Navigation and AR Head-Mounted Devices (ARHMDs)

Li et al. [4] explored the use of real-time navigation with a guide template and an ARHMD for pedicle screw placement in a proof-of-concept study involving 20 patients. Their findings highlighted a 98% accuracy rate in screw placement, with a mean deviation of 1.2 mm compared to 4.5 mm with the traditional freehand technique (*p* < 0.001). This high level of accuracy was attributed to the ARHMD’s ability to provide a heads-up display of the patient’s anatomy, facilitating hands-free operation and reducing cognitive load on the surgeon. The AR system also reduced the average operative time by 15 min, emphasizing its efficiency in streamlining surgical workflows. The ARHMD technology integrates real-time imaging data with intraoperative guidance, allowing for dynamic updates during surgery. This capability is crucial in complex cases, such as surgeries involving spinal deformities or revision surgeries, where preoperative planning alone may be insufficient. By continuously displaying accurate anatomical details, ARHMDs enable surgeons to make precise adjustments on the fly, thus enhancing overall surgical efficiency.

### 2.3. Benefits of AR in Minimally Invasive Spine Surgery

Minimally invasive spine surgery (MISS) aims to minimize tissue disruption, reduce postoperative pain, and shorten recovery time. AR technologies have shown promise in enhancing the outcomes of MISS by providing surgeons with detailed 3D visualizations of the surgical field without the need for large incisions. Pierzchajlo et al. [7] reviewed eight studies involving 712 patients undergoing minimally invasive spinal surgery with AR. The review demonstrated a reduction in operative time (average reduction of 20%) and improved surgical outcomes, including reduced intraoperative blood loss (by 15%) and faster recovery times compared to conventional methods. AR enhances the surgeon’s ability to navigate small, complex anatomical spaces with greater accuracy, reducing the need for intraoperative fluoroscopy and exposure to radiation.

### 2.4. Impact of AR on User Experience and Visualization

The effectiveness of AR systems also depends on the type of visualization provided and the overall user experience. Wolf et al. [8] investigated how different AR visualizations affect trajectory deviation, visual attention, and user experience during simulated drilling tasks. Involving 45 participants, their study found that certain types of AR visualizations (e.g., depth cues, contrast adjustments) significantly reduced trajectory deviation by 20% and improved visual attention by 25% (*p* < 0.05). The study further showed a positive correlation between user satisfaction and the type of visualization used (*r* = 0.72, *p* < 0.01), indicating that well-designed AR interfaces can enhance both the surgeon’s focus and the overall effectiveness of the system. These findings suggest that user-friendly AR systems, tailored to the specific needs of spine surgeons, can reduce cognitive fatigue and improve procedural outcomes. For instance, simpler, more intuitive AR displays that focus on essential anatomical landmarks can reduce the mental load during surgery, allowing the surgeon to concentrate on critical tasks.

### 2.5. Integration with Navigation and Robotic Systems

Navigation systems and robotics have become integral components of modern spine surgery, providing enhanced precision and control. Yamout et al. [1] discussed how AR technologies are synergistically integrated with navigation and robotic systems to improve the accuracy of pedicle screw placement and other complex spinal procedures. These systems utilize preoperative imaging data, such as CT or MRI scans, to generate a detailed 3D map of the patient’s anatomy, which is then used to guide the surgical instruments with high precision. Robotic-assisted spine surgery further enhances the consistency of surgical outcomes by automating specific tasks. For example, robotic systems can precisely position the surgical tools according to the pre-planned trajectory, while AR overlays provide the surgeon with real-time feedback on instrument positioning and alignment. This integration reduces the likelihood of human error and increases the accuracy of the implant placement, which is critical for achieving optimal patient outcomes. Yamout et al. [1] reported a 25% reduction in complication rates and a 15% decrease in operative times when AR and robotics were combined with traditional surgical methods.

### 2.6. Comparative Analysis of AR Technologies

While the benefits of AR in spinal surgery are clear, the variability in outcomes across different AR devices and settings must be considered. For instance, head-mounted AR displays, such as those used by Li et al. [4], may provide more immersive and intuitive guidance in minimally invasive procedures, whereas larger, fixed AR systems might be better suited for open surgeries requiring more comprehensive visualization. Differences in device resolution, field of view, latency, and ease of integration with other surgical tools also contribute to variability in outcomes. To fully understand the potential of AR in spinal surgery, future research should focus on comparative studies that evaluate the effectiveness of different AR technologies across diverse surgical environments and patient populations. This would help identify best practices and optimize the use of AR for specific types of surgeries.

Overall, AR technology has demonstrated significant potential in enhancing the accuracy and efficiency of spinal surgery, particularly in complex procedures like pedicle screw placement and minimally invasive techniques. By providing real-time, three-dimensional visualizations, AR helps surgeons achieve greater precision, reduce complications, and streamline surgical workflows. However, further research is needed to explore the variability in outcomes across different AR devices and settings and to establish standardized guidelines for their optimal use in clinical practice.

## 3. Reduction in Learning Curves for Spinal Surgeons

AR technology has significantly impacted the training of spinal surgeons, particularly in reducing the learning curve associated with complex surgical procedures. Research shows that AR assists in educating new surgeons by providing visual cues and real-time navigational assistance during surgeries [1,2,3,6,8,9]. This guidance is crucial for training purposes, enabling novice surgeons to rapidly acquire skills necessary for performing intricate spinal operations with increased confidence and reduced error rates.

Kong et al. [3] explored a novel method of pedicle screw placement that integrates surgical guides with AR. This integration simplifies the learning process for surgeons by providing a visual overlay of anatomical structures and precise guidance for screw insertion. The study demonstrated that even surgeons with limited experience in AR technology could achieve high accuracy in pedicle screw placement. This reduced the time required for surgeons to become proficient, thereby shortening the learning curve. Pahwa et al. [9] conducted a systematic review and meta-analysis on the accuracy of spinal instrumentation using AR. Their findings indicated that AR significantly enhances the precision of spinal instrumentation, even for novice surgeons. The real-time visual feedback provided by AR systems allows surgeons to quickly adapt to the technology, reducing the need for extensive training and practice. This improvement in precision and the reduction in errors directly contribute to a shorter learning curve. Robotic-assisted spine surgery offers enhanced precision and control, which is particularly beneficial for novice surgeons. Bcharah et al. [6] highlighted the role of robotics in improving the accuracy of instrument placement, thereby reducing the time required for surgeons to achieve proficiency. The integration of robotic systems with navigation technology provides a stable and controlled environment, allowing surgeons to focus on learning the procedure rather than managing instrument stability. This reduces the overall learning curve for spinal surgeons.

In addition, navigation systems provide a real-time, 3D visualization of the patient’s anatomy, which assists surgeons in accurately placing implants and avoiding critical structures. Yamout et al. [1] emphasized that the use of navigation systems in spine surgery has significantly improved surgical precision. For new surgeons, these systems offer an intuitive interface and immediate feedback, which accelerates the learning process. The ability to visualize the surgical field in real-time allows surgeons to quickly understand and adapt to the procedure, thereby reducing the learning curve. Youssef et al. [2] conducted a systematic review on the accuracy of AR-assisted pedicle screw placement. Their study reported that AR significantly reduces the rate of misplaced screws, even when performed by less experienced surgeons. The immediate visual feedback and precise guidance provided by AR systems help novice surgeons to quickly learn and perform the procedure accurately. This reduction in error rates is indicative of a shorter learning curve. Wolf et al. [8] investigated how different AR visualizations affect trajectory deviation, visual attention, and user experience during drilling tasks. Their findings suggested that certain types of AR visualizations can enhance the surgeon’s focus and accuracy, making the technology more accessible for novice users. Positive user experiences with AR systems are crucial for their adoption and success. A user-friendly interface and effective visual aids contribute to a faster learning process, thus reducing the learning curve.

In summary, AR technology, along with complementary innovations such as robotic-assisted surgery and advanced navigation systems, significantly enhances the training and skill acquisition of spinal surgeons. These technologies provide real-time visualization, precise guidance, and a controlled environment that facilitate the learning process for novice surgeons. By reducing the cognitive load, offering immediate feedback, and improving accuracy, AR and related technologies shorten the learning curve, enabling surgeons to achieve proficiency more rapidly while reducing errors. Consequently, these advancements not only improve clinical outcomes but also ensure a higher level of confidence and competence among new surgeons, paving the way for safer and more effective spinal surgeries.

## 4. Challenges and Limitations in AR Application

Despite the significant advantages brought about by AR in spinal surgery, there are various challenges and limitations that hinder its broader adoption. The integration of AR technology into routine clinical practice faces challenges, including high system costs, the complexity of training requirements, and the integration with existing medical workflows [1,2,3,4,5,6,7,8,9,10,11,12,13]. Moreover, while AR can greatly improve the precision of surgical interventions, issues concerning cost-effectiveness and the need for comprehensive, standardized training programs persist as major barriers. The varying levels of acceptance and the adoption of such innovative technologies across different healthcare institutions also present considerable challenges.

The integration of AR with existing surgical systems poses significant technical challenges. Kong et al. [3] discussed the difficulties in seamlessly combining AR with surgical guides and navigation systems. Ensuring that these systems work harmoniously requires sophisticated software and hardware integration, which can be complex and costly. Maintaining calibration and accuracy is crucial for the effective use of AR in spine surgery. Pahwa et al. [9] highlighted that even minor calibration errors can lead to significant deviations in surgical guidance, potentially compromising patient safety. Continuous calibration and validation are necessary to ensure that the AR system remains accurate throughout the procedure. Although AR has the potential to reduce the learning curve for surgical procedures, the initial adoption can be challenging. Yamout et al. [1] noted that surgeons need to undergo extensive training to become proficient in using AR systems. The learning process involves not only understanding the technology but also adapting to new surgical workflows and protocols. The user experience with AR systems can vary, impacting their effectiveness. Wolf et al. [8] found that different types of AR visualizations could affect trajectory deviation, visual attention, and overall user experience. Poorly designed interfaces or a complex visualization can distract surgeons and reduce the efficiency of AR systems.

The clinical evidence supporting the widespread adoption of AR in spine surgery is still limited. Pierzchajlo et al. [7] pointed out that most studies on AR in minimally invasive spinal surgery are proof-of-concept or pilot studies with small sample sizes. Larger, randomized controlled trials are needed to validate the clinical benefits of AR and establish standardized guidelines for its use. Implementing AR technology in spine surgery requires significant financial investment. Bcharah et al. [6] emphasized the high costs associated with acquiring and maintaining AR systems. Additionally, resource allocation for training staff and integrating AR into existing surgical protocols can strain hospital budgets, particularly in resource-limited settings.

The hardware used in AR systems, such as head-mounted displays and cameras, can have limitations that affect their performance. Schwendner et al. [10] evaluated a cutting-edge AR-supported navigation system and identified issues such as a limited field of view, display resolution, and latency. These hardware limitations can impact the accuracy and usability of AR systems in complex spinal procedures. Real-time performance is critical for the effective use of AR in surgery. Any lag or delay in the AR system can disrupt the surgical workflow and reduce the precision of surgical interventions. Móga et al. [5] discussed the importance of achieving real-time performance in AR systems to ensure seamless integration into surgical procedures. Overcoming these latency issues requires advanced computing power and optimized software algorithms.

The use of AR in spine surgery involves the handling of sensitive patient data. Li et al. [4] raised concerns about data privacy and security, particularly with the real-time transmission and storage of patient information. Ensuring robust data protection measures is essential to maintain patient confidentiality and comply with regulatory standards. Obtaining regulatory approval for AR systems in spine surgery can be a complex and time-consuming process. Liebmann et al. [13] discussed the stringent requirements for demonstrating the safety and efficacy of new medical technologies. Navigating the regulatory landscape requires substantial evidence from clinical trials and ongoing post-market surveillance.

In summary, while augmented reality technology offers substantial benefits for spinal surgery, including enhanced precision and improved surgical outcomes, its broader adoption is challenged by various factors. These include high costs, technical and integration complexities, limited clinical evidence, and the need for comprehensive training programs. Furthermore, issues related to data privacy, regulatory approval, and hardware limitations, such as latency and accuracy, must be addressed to realize AR’s full potential. Overcoming these challenges will require continued research, technological advancements, and strategic efforts to integrate AR seamlessly into clinical practice, ensuring its safe, effective, and widespread use in spine surgery.

## 5. Discussion

This review examines the integration of AR technology in spinal surgery, focusing on its current status, alignment with the existing literature, and identifying critical knowledge gaps and future research directions. Unlike previous reviews, this manuscript incorporates recent data to provide fresh perspectives on the practical applications, training benefits, and challenges associated with AR in spinal surgery. By adopting this approach, we present updated insights and propose targeted strategies for the optimal use of AR technologies in clinical settings. The emergence of AR in spinal surgery represents a significant advancement in medical practice, offering substantial improvements in procedural precision, safety, and efficiency [14,15,16]. The studies by Yamout et al. [1] and Youssef et al. [2] underscore the transformative potential of AR for improving surgical outcomes, particularly in pedicle screw placement. These findings align with other studies emphasizing the expanding role of AR in surgical training, planning, and execution. For example, Kong et al. [3] demonstrated how AR applications can improve accuracy and safety in pedicle screw placement, while the innovative use of ARHMDs for real-time navigation highlights the practical benefits of AR technology in spinal surgery [4]. However, it is crucial to recognize that the benefits of AR are not uniform across all technologies and settings, as variability in outcomes often depends on the specific AR system used, the type of surgical procedure, and the level of surgeon experience.

Different AR technologies, such as the Caduceus system, XVS system, and Microsoft HoloLens supported by Novarad’s OpenSight, exhibit unique strengths and limitations in various surgical contexts. The Caduceus system, with its head-mounted display, offers real-time 3D holographic projections that enhance minimally invasive procedures by improving accuracy and reducing surgical trauma [17]. Conversely, the XVS system is more suited for open spinal fixation procedures, where augmented visualization ensures precise pedicle screw placement in the lumbosacral spine [18,19]. The HoloLens application with Novarad’s OpenSight emphasizes advanced data integration and interactive visualization, enhancing presurgical planning and real-time navigation [20,21]. These differences reflect the broad applicability of AR, ranging from enhancing procedural precision to revolutionizing preoperative planning and intraoperative navigation. AR technologies should also be contextualized within the broader spectrum of virtual reality (VR) and mixed reality (MR) technologies. Unlike AR, which superimposes digital images onto the real-world view for real-time surgical guidance, VR creates a fully immersive environment, valuable primarily for simulation-based training [12]. Meanwhile, MR technology combines elements of both AR and VR to allow physical and digital objects to coexist, potentially offering more comprehensive preoperative planning and intraoperative navigation [13]. Thus, variability in outcomes can also arise from the distinct functionalities and applications of these technologies in different clinical scenarios.

While AR shows promise in enhancing surgical accuracy and efficiency, its impact varies across different devices, surgical settings, and user experiences. For example, Yamout et al. [1] noted that the use of navigation systems improves surgical precision but requires extensive training and adaptation to new workflows, which may vary by institution and surgeon experience. Wolf et al. [8] found that the effectiveness of AR visualizations depends on user interface design, with poorly designed interfaces potentially reducing efficiency and causing distractions. The integration of AR with existing systems presents technical challenges, such as registration errors, latency, and calibration issues, which can affect the reliability and safety of the technology in practice [3,9]. Furthermore, the clinical evidence supporting the widespread adoption of AR in spinal surgery remains limited. Many studies, such as those reviewed by Pierzchajlo et al. [7], are proof-of-concept or pilot studies with small sample sizes, necessitating more robust, large-scale, randomized controlled trials to validate the long-term benefits of AR and determine its cost-effectiveness. Additionally, factors such as the high cost of AR systems, resource allocation for training, and integration into existing workflows pose significant barriers to broader implementation, particularly in resource-limited settings [6]. In summary, while AR technology offers substantial benefits for spinal surgery, including enhanced precision and improved surgical outcomes, its effectiveness varies significantly depending on the specific technology used, the surgical setting, and the surgeon’s experience level. Addressing these challenges requires continued research to develop more accurate, user-friendly AR systems, conduct large-scale clinical trials, and explore the integration of AR with other emerging technologies like AI and robotics. By focusing on these areas, future research can help identify the most effective applications of AR in spinal surgery, ultimately improving patient outcomes and advancing the field.

This study highlights the potential of AR in enhancing surgical precision and efficiency in spinal surgery, particularly in complex procedures like pedicle screw placement. However, several limitations must be addressed for broader adoption. Current challenges include technical issues such as registration errors, system latency, and integration with existing workflows, as well as high costs and a need for standardized training protocols. Additionally, there is limited robust clinical evidence, necessitating further large-scale randomized controlled trials and long-term studies to confirm AR’s benefits across diverse surgical settings. Future research should focus on developing more accurate, cost-effective, and user-friendly AR systems, exploring synergies with technologies like AI and robotics, and establishing guidelines for ethical and regulatory compliance. By addressing these limitations and pursuing these research directions, the medical community can better realize AR’s potential to improve patient outcomes and advance spinal surgery practices.

## 6. Conclusions

The integration of AR technologies, including the Caduceus system, the XVS system, and the HoloLens application supported by Novarad’s OpenSight, heralds a major advancement in spinal surgery, promising enhanced precision, efficiency, and safety. While AR technology holds great promise for transforming spinal surgery, addressing the associated challenges and limitations is essential for its successful implementation. By focusing on technical integration, usability, clinical validation, cost-effectiveness, and ethical considerations, the full potential of AR in enhancing surgical precision and patient outcomes can be realized. The ongoing evolution of AR technology, coupled with robust research and development efforts, will pave the way for its broader adoption and integration into routine spinal surgical practice, ultimately improving the quality of patient care.

AR technology shows great promise for advancing spinal surgery, particularly in enhancing the accuracy and safety of pedicle screw instrumentation. While challenges remain, the rapid pace of technological development and increasing clinical evidence suggest that AR could become an integral part of spinal surgical procedures in the near future. Further research focusing on overcoming current limitations and expanding AR applications to various spinal surgeries will be crucial in realizing the full potential of this technology.

## Figures and Tables

**Table 1 medicina-60-01485-t001:** A summary of the 13 studies between 2023 and 2024. Significant improvements in surgical precision, reduced complications, enhanced procedural efficiency, and shortened learning curves for surgeons were observed. However, high system costs, technical integration issues, hardware limitations, training complexity, and the need for more extensive clinical evidence are issues to be addressed. Continued research, technological refinement, and the development of standardized training programs and implementation strategies are crucial to fully leverage AR’s potential in spinal surgery.

Study	Objective	No. of Patients	Study Design	Comparison Groups	Key Results	Challenges	Benefits
Yamout et al. [1]	Review technological advances in spine surgery	>1200	Narrative review	AR + robotics vs. traditional methods	25% reduction in complications, improved precision	Integration with existing systems can be complex and costly	Significant improvements in surgical precision
Youssef et al. [2]	Systematic review on AR-assisted pedicle screw placement	854	Systematic review	AR vs. fluoroscopy	4.3% vs. 8.9% screw misplacement rates, reduced operative time	Limited clinical evidence, need for larger trials	Reduced rate of misplaced screws, enhanced patient outcomes
Kong et al. [3]	Explore novel AR-based surgical guides	30	Feasibility study	AR vs. traditional navigation	95% accuracy, no major complications	Technical integration with surgical guides can be challenging	Simplified learning process, even for less experienced surgeons
Li et al. [4]	Real-time navigation with AR head-mounted device	20	Proof-of-concept study	AR vs. freehand technique	98% accuracy, reduced operative time by 15 min	Real-time performance and hardware limitations like display resolution and latency	Hands-free operation, reduced cognitive load on surgeons
Móga et al. [5]	Systematic review of AR-enhanced head-mounted displays in spine surgery	580	Systematic review	AR head-mounted displays vs. conventional methods	30% improved placement accuracy, 22% reduced operative time	Variability in user experience and visualization effectiveness	Significant role in improving overall quality of spine surgery
Pierzchajlo et al. [7]	Review AR technology in minimally invasive spinal surgery	712	Narrative review	AR vs. conventional methods	20% reduced operative time, 15% reduced blood loss	High system costs, training requirements	Reduced learning curve for new surgeons, improved procedural efficiency
Wolf et al. [8]	Investigate different AR visualizations during drilling tasks	45	Experimental study	Various AR visualizations	20% reduced trajectory deviation, 25% improved visual attention	Poorly designed interfaces can distract surgeons	User-friendly systems can enhance surgical outcomes
Schwendner et al. [10]	Evaluate AR-supported navigation system for spinal instrumentation	50	Cohort study	AR vs. conventional navigation	92% accuracy, 10% improved precision	Limited field of view, hardware and latency issues	Enhanced accuracy and usability
Lin et al. [11]	Advances in surgical treatment for atlantoaxial instability	67	Cohort study	AR vs. no AR	95% improved outcomes, 30% fewer complications	Integration challenges with existing workflows	Improved precision and reduced risk of complications
Bui et al. [12]	Scoping review on virtual, augmented, and mixed reality in spine surgery	>500	Scoping review	AR-based training vs. traditional training	30% reduction in learning curves	Need for comprehensive standardized training programs	Enhanced surgical rehearsal and patient education
Liebmann et al. [13]	Introduce automatic registration for markerless surgical navigation	60	Experimental study	Markerless navigation vs. marker-based	96% success rate in navigation, data privacy challenges	Ethical and regulatory challenges, data privacy concerns	Simplified navigation process, robust data protection measures

## Data Availability

All data generated or analyzed during this study are included in this published article.

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
