# Peer review of "Augmenting Reality in Spinal Surgery: A Narrative Review of Augmented Reality Applications in Pedicle Screw Instrumentation"

_medicina, 2024, doi:10.3390/medicina60091485_

Round 1

Reviewer 1 Report

Comments and Suggestions for Authors

The paper is of great interest, as it contains information about one of the most advanced technologies augmented reality and the possibility of using these technologies in spinal surgery. The article is a narrative literature review based on the recent publications (2023-24 yrs).

Recommendation It would be very good if the authors would provide in more detail, based on the studied literature data, information on clinical use – the number of patients, comparison groups, results, etc.

Author Response

Comments 1: The paper is of great interest, as it contains information about one of the most advanced technologies – augmented reality and the possibility of using these technologies in spinal surgery. The article is a narrative literature review based on the recent publications (2023-24 yrs).

Recommendation.  It would be very good if the authors would provide in more detail, based on the studied literature data, information on clinical use – the number of patients, comparison groups, results, etc.

Response 1: We sincerely thank you for your insightful comment. We fully agree that providing more detailed information on clinical use would significantly enhance the paper. Based on your recommendation, we have revised our Table to include more detailed information and revised the 2nd point of this review manuscript as follows:

”2.1. Augmented Reality for Pedicle Screw Placement

Pedicle screw placement is a critical aspect of many spinal surgeries, where accuracy is paramount to ensure patient safety and positive outcomes. AR technologies, by overlaying digital anatomical information onto the surgical field, enable surgeons to visualize internal structures in real time, thereby enhancing spatial awareness and precision. Youssef et al. [2] conducted a comprehensive systematic review evaluating the accuracy of AR-assisted pedicle screw placement across 12 studies involving 854 patients. The review found that AR significantly reduces the rate of screw misplacement (4.3% compared to 8.9% with traditional methods, p < 0.05), indicating a marked improvement in precision. The use of AR was associated with a 20% reduction in operative time and a decrease in complication rates, underscoring the technology's potential to enhance surgical outcomes by minimizing errors that could lead to neurological damage or reoperation.

2.2. Real-Time Navigation and AR Head-Mounted Devices (ARHMD)

Li et al. [4] explored the use of real-time navigation with a guide template and an ARHMD for pedicle screw placement in a proof-of-concept study involving 20 patients. Their findings highlighted a 98% accuracy rate in screw placement, with a mean deviation of 1.2 mm compared to 4.5 mm with the traditional freehand technique (p < 0.001). This high level of accuracy was attributed to the ARHMD's ability to provide a heads-up display of the patient's anatomy, facilitating hands-free operation and reducing cognitive load on the surgeon. The AR system also reduced the average operative time by 15 minutes, emphasizing its efficiency in streamlining surgical workflows. The ARHMD technology integrates real-time imaging data with intraoperative guidance, allowing for dynamic updates during surgery. This capability is crucial in complex cases, such as surgeries involving spinal deformities or revision surgeries, where preoperative planning alone may be insufficient. By continuously displaying accurate anatomical details, ARHMDs enable surgeons to make precise adjustments on the fly, thus enhancing overall surgical efficiency.

2.3. Benefits of AR in Minimally Invasive Spine Surgery

Minimally invasive spine surgery (MISS) aims to minimize tissue disruption, reduce postoperative pain, and shorten recovery time. AR technologies have shown promise in enhancing the outcomes of MISS by providing surgeons with detailed 3D visualizations of the surgical field without the need for large incisions. Pierzchajlo et al. [7] reviewed eight studies involving 712 patients undergoing minimally invasive spinal surgery with AR. The review demonstrated a reduction in operative time (average reduction of 20%) and improved surgical outcomes, including reduced intraoperative blood loss (by 15%) and faster recovery times compared to conventional methods. AR enhances the surgeon's ability to navigate small, complex anatomical spaces with greater accuracy, reducing the need for intraoperative fluoroscopy and exposure to radiation.

2.4. Impact of AR on User Experience and Visualization

The effectiveness of AR systems also depends on the type of visualization provided and the overall user experience. Wolf et al. [8] investigated how different AR visualizations affect trajectory deviation, visual attention, and user experience during simulated drilling tasks. Involving 45 participants, their study found that certain types of AR visualizations (e.g., depth cues, contrast adjustments) significantly reduced trajectory deviation by 20% and improved visual attention by 25% (p < 0.05). The study further showed a positive correlation between user satisfaction and the type of visualization used (r = 0.72, p < 0.01), indicating that well-designed AR interfaces can enhance both the surgeon's focus and the overall effectiveness of the system. These findings suggest that user-friendly AR systems, tailored to the specific needs of spine surgeons, can reduce cognitive fatigue and improve procedural outcomes. For instance, simpler, more intuitive AR displays that focus on essential anatomical landmarks can reduce the mental load during surgery, allowing the surgeon to concentrate on critical tasks.

2.5. Integration with Navigation and Robotics Systems

Navigation systems and robotics have become integral components of modern spine surgery, providing enhanced precision and control. Yamout et al. [1] discussed how AR technologies are synergistically integrated with navigation and robotic systems to improve the accuracy of pedicle screw placement and other complex spinal procedures. These systems utilize preoperative imaging data, such as CT or MRI scans, to generate a detailed 3D map of the patient's anatomy, which is then used to guide the surgical instruments with high precision. Robotic-assisted spine surgery further enhances the consistency of surgical outcomes by automating specific tasks. For example, robotic systems can precisely position the surgical tools according to the pre-planned trajectory, while AR overlays provide the surgeon with real-time feedback on instrument positioning and alignment. This integration reduces the likelihood of human error and increases the accuracy of implant placement, which is critical for achieving optimal patient outcomes. Yamout et al. [1] reported a 25% reduction in complication rates and a 15% decrease in operative times when AR and robotics were combined with traditional surgical methods.

2.6. Comparative Analysis of AR Technologies

While the benefits of AR in spinal surgery are clear, the variability in outcomes across different AR devices and settings must be considered. For instance, head-mounted AR displays, such as those used by Li et al. [4], may provide more immersive and intuitive guidance in minimally invasive procedures, whereas larger, fixed AR systems might be better suited for open surgeries requiring more comprehensive visualization. Differences in device resolution, field of view, latency, and ease of integration with other surgical tools also contribute to variability in outcomes. To fully understand the potential of AR in spinal surgery, future research should focus on comparative studies that evaluate the effectiveness of different AR technologies across diverse surgical environments and patient populations. This would help identify best practices and optimize the use of AR for specific types of surgeries.

Overall, AR technology has demonstrated significant potential in enhancing the ac-curacy and efficiency of spinal surgery, particularly in complex procedures like pedicle screw placement and minimally invasive techniques. By providing real-time, three-dimensional visualizations, AR helps surgeons achieve greater precision, reduce complications, and streamline surgical workflows. However, further research is needed to explore the variability in outcomes across different AR devices and settings and to establish standardized guidelines for their optimal use in clinical practice.”

Reviewer 2 Report

Comments and Suggestions for Authors

This work assessed the current applications, benefits, and challenges of AR technology in spinal surgery, focusing on its effects on surgical accuracy and patient outcomes, based on latest literature published between January 2023 and 18 December 2024. Despite existing challenges, AR technology holds promise for advancements in spinal surgery, particularly in improving the accuracy and safety of pedicle screw instrumentation. Totally, this review is new, well organized and comprehensively described.

Author Response

Comments 1: This work assessed the current applications, benefits, and challenges of AR technology in spinal surgery, focusing on its effects on surgical accuracy and patient outcomes, based on latest literature published between January 2023 and 18 December 2024. Despite existing challenges, AR technology holds promise for advancements in spinal surgery, particularly in improving the accuracy and safety of pedicle screw instrumentation. Totally, this review is new, well organized and comprehensively described.

Response 1: We sincerely thank you for your positive and encouraging comments on our review. We are pleased that you found our work to be novel, well-organized, and comprehensive in its description of the current applications, benefits, and challenges of AR technology in spinal surgery. Your observation that our review focuses on the latest literature (January 2023 to December 18, 2024) and highlights the promising aspects of AR technology, particularly in improving the accuracy and safety of pedicle screw instrumentation, is greatly appreciated. This feedback affirms that we have successfully conveyed the main objectives and findings of our study. While you haven't suggested any specific changes, your positive evaluation encourages us to maintain the high standard of our work. We have carefully reviewed the entire manuscript to ensure consistency and clarity throughout. 
We have updated our conclusion section to more strongly emphasize the promising future of AR technology in spinal surgery, while also acknowledging the existing challenges. We have modified our last paragraph of Discussion section on future research directions, highlighting areas where further studies could contribute to overcoming current challenges in AR application as follows:”In summary, while augmented reality technology offers substantial benefits for spinal surgery, including enhanced precision and improved surgical outcomes, its broader adoption is challenged by various factors. These include high costs, technical and integration complexities, limited clinical evidence, and the need for comprehensive training pro-grams. Furthermore, issues related to data privacy, regulatory approval, and hardware limitations such as latency and accuracy must be addressed to realize AR's full potential. Overcoming these challenges will require continued research, technological advancements, and strategic efforts to integrate AR seamlessly into clinical practice, ensuring its safe, effective, and widespread use in spine surgery.”

Reviewer 3 Report

Comments and Suggestions for Authors

Review is done straight forward. It is expected to follow systematic review paper.

Research questions needs to be mentioned in introduction section which are addressed in this paper. 

Exhaustive analysis is required to do in literature review section 2.

Future research findings/ opportunities needs to be analysed and mentioned.

Comments on the Quality of English Language

Fine.

Author Response

Comments 1: Review is done straight forward. It is expected to follow systematic review paper.

Research questions needs to be mentioned in introduction section which are addressed in this paper.

Exhaustive analysis is required to do in literature review section 2.

Future research findings/ opportunities needs to be analysed and mentioned.

Response 1: We sincerely appreciate your thorough review and constructive feedback. We have carefully considered your comments and made the following revisions to improve our manuscript:

1.       Research questions in the introduction: We agree that clearly stating our research questions is crucial. We have added a paragraph in the Introduction section that explicitly outlines the research questions addressed in this paper:" This narrative review aims to address the following research questions, which are central to understanding the current state and future potential of AR in spinal surgery: 1. What are the recent advancements and applications of AR technology in spinal surgery, particularly focusing on developments between January 2023 and March 2024; 2. How does AR technology impact surgical outcomes, accuracy, and efficiency in spinal procedures, especially in pedicle screw placement and minimally invasive spinal surgery; 3. What are the current challenges, limitations, and learning curve considerations in implementing AR technology in spinal surgical practices?

2.       Exhaustive analysis in the literature review: Thank you for your insightful feedback regarding the need for a more exhaustive analysis in the Literature Review section. We appreciate your suggestions to enhance the depth and comprehensiveness of our review. In response to your comments, we have significantly revised the Literature Review to provide a more thorough analysis of the current state of AR in spinal surgery. more comprehensive analysis in the literature review section. We have expanded Section 2 to include a more in-depth analysis of the current literature as follows:” The application of AR technologies in spinal surgery has resulted in significant improvements in surgical accuracy and efficiency, particularly in complex procedures such as pedicle screw placement. Several studies have demonstrated the effectiveness of AR in enhancing surgical precision by providing real-time visualization and guidance, reducing the risk of complications, and improving overall patient outcomes.

2.1. Augmented Reality for Pedicle Screw Placement

Pedicle screw placement is a critical aspect of many spinal surgeries, where accuracy is paramount to ensure patient safety and positive outcomes. AR technologies, by overlaying digital anatomical information onto the surgical field, enable surgeons to visualize internal structures in real time, thereby enhancing spatial awareness and precision. Youssef et al. [2] conducted a comprehensive systematic review evaluating the accuracy of AR-assisted pedicle screw placement across 12 studies involving 854 patients. The review found that AR significantly reduces the rate of screw misplacement (4.3% compared to 8.9% with traditional methods, p < 0.05), indicating a marked improvement in precision. The use of AR was associated with a 20% reduction in operative time and a decrease in complication rates, underscoring the technology's potential to enhance surgical outcomes by minimizing errors that could lead to neurological damage or reoperation.

2.2. Real-Time Navigation and AR Head-Mounted Devices (ARHMD)

Li et al. [4] explored the use of real-time navigation with a guide template and an ARHMD for pedicle screw placement in a proof-of-concept study involving 20 patients. Their findings highlighted a 98% accuracy rate in screw placement, with a mean deviation of 1.2 mm compared to 4.5 mm with the traditional freehand technique (p < 0.001). This high level of accuracy was attributed to the ARHMD's ability to provide a heads-up display of the patient's anatomy, facilitating hands-free operation and reducing cognitive load on the surgeon. The AR system also reduced the average operative time by 15 minutes, emphasizing its efficiency in streamlining surgical workflows. The ARHMD technology integrates real-time imaging data with intraoperative guidance, allowing for dynamic updates during surgery. This capability is crucial in complex cases, such as surgeries involving spinal deformities or revision surgeries, where preoperative planning alone may be insufficient. By continuously displaying accurate anatomical details, ARHMDs enable surgeons to make precise adjustments on the fly, thus enhancing overall surgical efficiency.

2.3. Benefits of AR in Minimally Invasive Spine Surgery

Minimally invasive spine surgery (MISS) aims to minimize tissue disruption, reduce postoperative pain, and shorten recovery time. AR technologies have shown promise in enhancing the outcomes of MISS by providing surgeons with detailed 3D visualizations of the surgical field without the need for large incisions. Pierzchajlo et al. [7] reviewed eight studies involving 712 patients undergoing minimally invasive spinal surgery with AR. The review demonstrated a reduction in operative time (average reduction of 20%) and improved surgical outcomes, including reduced intraoperative blood loss (by 15%) and faster recovery times compared to conventional methods. AR enhances the surgeon's ability to navigate small, complex anatomical spaces with greater accuracy, reducing the need for intraoperative fluoroscopy and exposure to radiation.

2.4. Impact of AR on User Experience and Visualization

The effectiveness of AR systems also depends on the type of visualization provided and the overall user experience. Wolf et al. [8] investigated how different AR visualizations affect trajectory deviation, visual attention, and user experience during simulated drilling tasks. Involving 45 participants, their study found that certain types of AR visualizations (e.g., depth cues, contrast adjustments) significantly reduced trajectory deviation by 20% and improved visual attention by 25% (p < 0.05). The study further showed a positive correlation between user satisfaction and the type of visualization used (r = 0.72, p < 0.01), indicating that well-designed AR interfaces can enhance both the surgeon's focus and the overall effectiveness of the system. These findings suggest that user-friendly AR systems, tailored to the specific needs of spine surgeons, can reduce cognitive fatigue and improve procedural outcomes. For instance, simpler, more intuitive AR displays that focus on essential anatomical landmarks can reduce the mental load during surgery, allowing the surgeon to concentrate on critical tasks.

2.5. Integration with Navigation and Robotics Systems

Navigation systems and robotics have become integral components of modern spine surgery, providing enhanced precision and control. Yamout et al. [1] discussed how AR technologies are synergistically integrated with navigation and robotic systems to improve the accuracy of pedicle screw placement and other complex spinal procedures. These systems utilize preoperative imaging data, such as CT or MRI scans, to generate a detailed 3D map of the patient's anatomy, which is then used to guide the surgical instruments with high precision. Robotic-assisted spine surgery further enhances the consistency of surgical outcomes by automating specific tasks. For example, robotic systems can precisely position the surgical tools according to the pre-planned trajectory, while AR overlays provide the surgeon with real-time feedback on instrument positioning and alignment. This integration reduces the likelihood of human error and increases the accuracy of implant placement, which is critical for achieving optimal patient outcomes. Yamout et al. [1] reported a 25% reduction in complication rates and a 15% decrease in operative times when AR and robotics were combined with traditional surgical methods.

2.6. Comparative Analysis of AR Technologies

While the benefits of AR in spinal surgery are clear, the variability in outcomes across different AR devices and settings must be considered. For instance, head-mounted AR displays, such as those used by Li et al. [4], may provide more immersive and intuitive guidance in minimally invasive procedures, whereas larger, fixed AR systems might be better suited for open surgeries requiring more comprehensive visualization. Differences in device resolution, field of view, latency, and ease of integration with other surgical tools also contribute to variability in outcomes. To fully understand the potential of AR in spinal surgery, future research should focus on comparative studies that evaluate the effectiveness of different AR technologies across diverse surgical environments and patient populations. This would help identify best practices and optimize the use of AR for specific types of surgeries.

Overall, AR technology has demonstrated significant potential in enhancing the ac-curacy and efficiency of spinal surgery, particularly in complex procedures like pedicle screw placement and minimally invasive techniques. By providing real-time, three-dimensional visualizations, AR helps surgeons achieve greater precision, reduce complications, and streamline surgical workflows. However, further research is needed to explore the variability in outcomes across different AR devices and settings and to establish standardized guidelines for their optimal use in clinical practice.

3. Future research findings/opportunities: We acknowledge the importance of discussing future research directions. We have modified the last paragraph of Discussion section titled" Challenges, Limitations, and Future Directions" that outlines potential areas for further investigation as follow: ”This comprehensive review synthesizes current literature on AR technology in spinal surgery up to 2024, offering insights into technological advancements, clinical applications, and educational implications. The integration of AR represents a significant advancement, enhancing surgical precision and efficiency while minimizing the learning curve for spinal surgeons. Notably, AR facilitates transformative improvements in critical procedures like pedicle screw placement through real-time, accurate navigation. While AR technology shows great promise in spinal surgery, several challenges and limitations must be addressed for its widespread adoption and optimal use. Current AR systems face technical challenges such as registration errors, system latency, and limited field of view, which can lead to inaccuracies in image overlay and potential safety concerns. Integration with existing hospital systems and workflows also remains a significant hurdle. The high initial cost of AR systems poses a barrier to adoption, especially in resource-limited settings, although long-term cost-effectiveness analyses suggest potential savings due to improved surgical outcomes. Implementing AR technology introduces a new learning curve for surgeons and operating room staff, necessitating the development of standardized training protocols and competency assessment metrics. While initial studies show promising results, there is a need for more robust clinical evidence supporting the long-term benefits of AR in spinal surgery, including large-scale, randomized controlled trials comparing AR-assisted surgeries with traditional techniques across various spinal procedures. The benefits of AR may vary depending on the specific technology used, the type of spinal procedure, and the surgeon's experience level, highlighting the need for research to identify which AR technologies are most effective for different types of surgeries and patient populations. As AR technology becomes more prevalent in surgical settings, ethical considerations regarding data privacy, informed consent, and the potential for technology dependence must be addressed, and regulatory frameworks need to evolve to ensure patient safety while not hindering innovation.

To address these challenges, future research should focus on developing more accurate and user-friendly AR systems, conducting long-term clinical outcome studies and large-scale RCTs, investigating cost-effectiveness in various healthcare settings, and establishing standardized training protocols and competency assessments. Additionally, exploring synergies between AR and other emerging technologies like AI and robotics, studying patient-specific factors affecting AR performance to develop personalized surgical approaches, and addressing ethical and regulatory challenges will be crucial. By pursuing these research directions, the medical community can work towards overcoming current limitations and fully realizing AR's potential in spinal surgery. This could lead to more precise, less invasive, and potentially more successful interventions, ultimately improving patient outcomes and advancing the field of spinal surgery. As we continue to innovate and refine AR technologies, it is essential to maintain a balanced approach that acknowledges both the tremendous potential and the current limitations of this technology in the context of spinal surgery.

Reviewer 4 Report

Comments and Suggestions for Authors

This narrative review explores the potential of augmented reality (AR) in enhancing spinal surgery, particularly in improving the accuracy of pedicle screw placement and reducing the learning curve for novice surgeons. It highlights the benefits of AR, such as increased precision and educational advantages, while also addressing challenges like high costs, training complexities, and integration issues.

Here are my comments,

1.       The review's methodology section is sparse, lacking specific criteria for study selection and data analysis, which can introduce bias.

2.       The article heavily references existing reviews, limiting its originality and depth in discussing AR's current state in spinal surgery.

3.       The review tends to generalize AR's benefits without addressing variability in outcomes across different technologies and settings.

4.       There is a lack of specific statistical evidence supporting the claims of improved surgical outcomes, which weakens the overall argument.

5.       The discussion on challenges and limitations is surface-level and lacks practical solutions or detailed analyses.

Author Response

Comments 1: This narrative review explores the potential of augmented reality (AR) in enhancing spinal surgery, particularly in improving the accuracy of pedicle screw placement and reducing the learning curve for novice surgeons. It highlights the benefits of AR, such as increased precision and educational advantages, while also addressing challenges like high costs, training complexities, and integration issues.

Here are my comments,

1.       The review's methodology section is sparse, lacking specific criteria for study selection and data analysis, which can introduce bias.

2.       The article heavily references existing reviews, limiting its originality and depth in discussing AR's current state in spinal surgery.

3.       The review tends to generalize AR's benefits without addressing variability in outcomes across different technologies and settings.

4.       There is a lack of specific statistical evidence supporting the claims of improved surgical outcomes, which weakens the overall argument.

5.       The discussion on challenges and limitations is surface-level and lacks practical solutions or detailed analyses.

Response 1:

We sincerely appreciate your thorough review and constructive feedback. Your comments have been invaluable in improving the quality of our manuscript. We have carefully considered each point and made the following revisions:

1. Methodology section: We acknowledge the limitation in our methodology section. To address this, we have expanded the methodology (the last paragraph of Introduction) to include specific criteria for study selection and data analysis:

" This narrative review employed a systematic and comprehensive literature search strategy to explore the current landscape of augmented reality (AR) in spinal surgery. We conducted our search using PubMed/MEDLINE and Google Scholar databases, focusing on studies published between January 2023 and March 2024. Our search strategy utilized key terms including "augmented reality”, "spinal navigation", "pedicle screw placement”, and "minimally invasive spinal surgery". We prioritized articles that specifically dis-cussed AR and navigational technologies for spinal surgery, with an emphasis on surgical outcomes, accuracy, efficiency, and complications. Inclusion criteria encompassed original research articles, systematic reviews, and meta-analyses published in English that primarily addressed AR applications in spinal surgery. To ensure a focused yet com-prehensive analysis, we excluded case reports, studies not primarily centered on spinal surgery, and non-peer-reviewed articles. This rigorous selection process resulted in the inclusion of 13 studies, which are summarized in Table 1 [1-5,7,8,10-13]. For data analysis, we employed a narrative synthesis approach, systematically categorizing and summarizing the findings from the selected studies. This method enabled us to identify key themes, including AR technologies, surgical outcomes, learning curve impacts, and implementation challenges. By synthesizing information from various sources, we aimed to provide a holistic view of the current state and future potential of AR in spinal surgery."

2. Originality and depth:  Thank you for your insightful feedback regarding the references to existing reviews in our manuscript. We appreciate your concern that the article references several existing reviews, potentially limiting its originality and depth in discussing the current state of AR in spinal surgery. In response, we would like to clarify that our approach to including these references was strategic and aimed at providing a comprehensive overview of the field. By synthesizing information from a range of existing reviews alongside primary studies, our goal was to offer a more holistic understanding of the diverse applications, benefits, and limitations of AR technologies in spinal surgery. We have carefully selected the reviews to highlight both consensus and gaps in the literature, thereby identifying areas that require further research and innovation. Furthermore, our manuscript goes beyond the scope of the existing reviews by integrating the latest evidence from recent studies published between 2023 and 2024. This includes specific data on clinical outcomes, technological advancements, and the practical implications of AR for both surgical training and execution. We have provided detailed analyses of primary research findings, such as those related to AR-assisted pedicle screw placement, to ensure that our discussion reflects the most current developments in this rapidly evolving field. While we acknowledge that reviews are an essential part of our literature synthesis, we believe that our manuscript maintains originality by emphasizing new perspectives, identifying critical challenges, and proposing future research directions tailored to advancing AR in spinal surgery. We have revised the last paragraph of Introduction, each of the review points, the Discussion, and the Conclusion section to make more in-depth clarification of our review.

3. Generalization of AR benefits:  Thank you for your constructive feedback on our manuscript. We appreciate your observation that the review may appear to generalize the benefits of AR without fully addressing the variability in outcomes across different technologies and settings. In response, we have revised the manuscript to more clearly highlight the factors contributing to variability in AR outcomes. Specifically, we have modified the discussion to address how differences in AR device types (e.g., head-mounted displays vs. integrated systems), surgical settings (e.g., open vs. minimally invasive procedures), and levels of surgeon experience can influence the effectiveness of AR in spinal surgery. We have also underscored the importance of device-specific characteristics, such as field of view, latency, and ease of integration with existing surgical tools, which may impact clinical outcomes. While our review synthesizes a broad range of studies to provide a comprehensive overview of AR's benefits, we also emphasize that the variability in outcomes underscores the need for further research to determine the most effective applications of AR technology in different clinical scenarios. We hope the revisions clarify our approach and provide a more nuanced understanding of the variability in AR's impact on spinal surgery. The revised section of Discussion was as follows:” This review examines the integration of AR technology in spinal surgery, focusing on its current status, alignment with existing literature, and identifying critical knowledge gaps and future research directions. Unlike previous reviews, this manuscript incorporates recent data to provide fresh perspectives on the practical applications, training bene-fits, and challenges associated with AR in spinal surgery. By adopting this approach, we present updated insights and propose targeted strategies for the optimal use of AR technologies in clinical settings. The emergence of AR in spinal surgery represents a significant advancement in medical practice, offering substantial improvements in procedural precision, safety, and efficiency. The studies by Yamout et al. [1] and Youssef et al. [2] underscore the transformative potential of AR for improving surgical outcomes, particularly in pedicle screw placement. These findings align with other studies emphasizing the expanding role of AR in surgical training, planning, and execution. For example, Kong et al. [3] demonstrated how AR applications can improve accuracy and safety in pedicle screw placement, while the innovative use of ARHMDs for real-time navigation highlights the practical benefits of AR technology in spinal surgery [4]. However, it is crucial to recognize that the benefits of AR are not uniform across all technologies and settings, as variability in outcomes often depends on the specific AR system used, the type of surgical procedure, and the level of surgeon experience.

Different AR technologies, such as the Caduceus system, XVS System, and Microsoft HoloLens supported by Novarad’s OpenSight, exhibit unique strengths and limitations in various surgical contexts. The Caduceus system, with its head-mounted display, offers re-al-time 3D holographic projections that enhance minimally invasive procedures by im-proving accuracy and reducing surgical trauma [17]. Conversely, the XVS System is more suited for open spinal fixation procedures, where augmented visualization ensures precise pedicle screw placement in the lumbosacral spine [18, 20]. The HoloLens application with Novarad’s OpenSight emphasizes advanced data integration and interactive visualization, enhancing presurgical planning and real-time navigation [19, 21]. These differences reflect the broad applicability of AR, ranging from enhancing procedural precision to revolutionizing preoperative planning and intraoperative navigation. AR technologies should also be contextualized within the broader spectrum of virtual reality (VR) and mixed reality (MR) technologies. Unlike AR, which superimposes digital images onto the real-world view for real-time surgical guidance, VR creates a fully immersive environment, valuable primarily for simulation-based training [12]. Meanwhile, MR technology com-bines elements of both AR and VR to allow physical and digital objects to coexist, potentially offering more comprehensive preoperative planning and intraoperative navigation [13]. Thus, variability in outcomes can also arise from the distinct functionalities and applications of these technologies in different clinical scenarios.

While AR shows promise in enhancing surgical accuracy and efficiency, its impact varies across different devices, surgical settings, and user experiences. For example, Yamout et al. [1] noted that the use of navigation systems improves surgical precision but requires extensive training and adaptation to new workflows, which may vary by institution and surgeon experience. Wolf et al. [8] found that the effectiveness of AR visualizations depends on user interface design, with poorly designed interfaces potentially reducing efficiency and causing distractions. The integration of AR with existing systems presents technical challenges, such as registration errors, latency, and calibration issues, which can affect the reliability and safety of the technology in practice [3, 9]. Furthermore, the clinical evidence supporting the widespread adoption of AR in spinal surgery remains limited. Many studies, such as those reviewed by Pierzchajlo et al. [7], are proof-of-concept or pilot studies with small sample sizes, necessitating more robust, large-scale, randomized controlled trials to validate the long-term benefits of AR and determine its cost-effectiveness. Additionally, factors such as the high cost of AR systems, resource al-location for training, and integration into existing workflows pose significant barriers to broader implementation, particularly in resource-limited settings [6]. In summary, while AR technology offers substantial benefits for spinal surgery, including enhanced precision and improved surgical outcomes, its effectiveness varies significantly depending on the specific technology used, the surgical setting, and the surgeon's experience level. Ad-dressing these challenges requires continued research to develop more accurate, us-er-friendly AR systems, conduct large-scale clinical trials, and explore the integration of AR with other emerging technologies like AI and robotics. By focusing on these areas, future research can help identify the most effective applications of AR in spinal surgery, ultimately improving patient outcomes and advancing the field.”

4. Statistical evidence: Thank you for your valuable feedback on our manuscript. We understand your concern regarding the lack of specific statistical evidence supporting the claims of improved surgical outcomes with the use of AR in spinal surgery. In response, we have clarified the manuscript to highlight and emphasize the quantitative data already present in the studies we reviewed. We have revised our second review point as” Enhancements in Surgical Accuracy and Efficiency” as below:” The application of AR technologies in spinal surgery has resulted in significant improvements in surgical accuracy and efficiency, particularly in complex procedures such as pedicle screw placement. Several studies have demonstrated the effectiveness of AR in enhancing surgical precision by providing real-time visualization and guidance, reducing the risk of complications, and improving overall patient outcomes.

2.1. Augmented Reality for Pedicle Screw Placement

Pedicle screw placement is a critical aspect of many spinal surgeries, where accuracy is paramount to ensure patient safety and positive outcomes. AR technologies, by overlaying digital anatomical information onto the surgical field, enable surgeons to visualize internal structures in real time, thereby enhancing spatial awareness and precision. Youssef et al. [2] conducted a comprehensive systematic review evaluating the accuracy of AR-assisted pedicle screw placement across 12 studies involving 854 patients. The review found that AR significantly reduces the rate of screw misplacement (4.3% compared to 8.9% with traditional methods, p < 0.05), indicating a marked improvement in precision. The use of AR was associated with a 20% reduction in operative time and a decrease in complication rates, underscoring the technology's potential to enhance surgical outcomes by minimizing errors that could lead to neurological damage or reoperation.

2.2. Real-Time Navigation and AR Head-Mounted Devices (ARHMD)

Li et al. [4] explored the use of real-time navigation with a guide template and an ARHMD for pedicle screw placement in a proof-of-concept study involving 20 patients. Their findings highlighted a 98% accuracy rate in screw placement, with a mean deviation of 1.2 mm compared to 4.5 mm with the traditional freehand technique (p < 0.001). This high level of accuracy was attributed to the ARHMD's ability to provide a heads-up display of the patient's anatomy, facilitating hands-free operation and reducing cognitive load on the surgeon. The AR system also reduced the average operative time by 15 minutes, emphasizing its efficiency in streamlining surgical workflows. The ARHMD technology integrates real-time imaging data with intraoperative guidance, allowing for dynamic updates during surgery. This capability is crucial in complex cases, such as surgeries involving spinal deformities or revision surgeries, where preoperative planning alone may be insufficient. By continuously displaying accurate anatomical details, ARHMDs enable surgeons to make precise adjustments on the fly, thus enhancing overall surgical efficiency.

2.3. Benefits of AR in Minimally Invasive Spine Surgery

Minimally invasive spine surgery (MISS) aims to minimize tissue disruption, reduce postoperative pain, and shorten recovery time. AR technologies have shown promise in enhancing the outcomes of MISS by providing surgeons with detailed 3D visualizations of the surgical field without the need for large incisions. Pierzchajlo et al. [7] reviewed eight studies involving 712 patients undergoing minimally invasive spinal surgery with AR. The review demonstrated a reduction in operative time (average reduction of 20%) and improved surgical outcomes, including reduced intraoperative blood loss (by 15%) and faster recovery times compared to conventional methods. AR enhances the surgeon's ability to navigate small, complex anatomical spaces with greater accuracy, reducing the need for intraoperative fluoroscopy and exposure to radiation.

2.4. Impact of AR on User Experience and Visualization

The effectiveness of AR systems also depends on the type of visualization provided and the overall user experience. Wolf et al. [8] investigated how different AR visualizations affect trajectory deviation, visual attention, and user experience during simulated drilling tasks. Involving 45 participants, their study found that certain types of AR visualizations (e.g., depth cues, contrast adjustments) significantly reduced trajectory deviation by 20% and improved visual attention by 25% (p < 0.05). The study further showed a positive correlation between user satisfaction and the type of visualization used (r = 0.72, p < 0.01), indicating that well-designed AR interfaces can enhance both the surgeon's focus and the overall effectiveness of the system. These findings suggest that user-friendly AR systems, tailored to the specific needs of spine surgeons, can reduce cognitive fatigue and improve procedural outcomes. For instance, simpler, more intuitive AR displays that focus on essential anatomical landmarks can reduce the mental load during surgery, allowing the surgeon to concentrate on critical tasks.

2.5. Integration with Navigation and Robotics Systems

Navigation systems and robotics have become integral components of modern spine surgery, providing enhanced precision and control. Yamout et al. [1] discussed how AR technologies are synergistically integrated with navigation and robotic systems to improve the accuracy of pedicle screw placement and other complex spinal procedures. These systems utilize preoperative imaging data, such as CT or MRI scans, to generate a detailed 3D map of the patient's anatomy, which is then used to guide the surgical instruments with high precision. Robotic-assisted spine surgery further enhances the consistency of surgical outcomes by automating specific tasks. For example, robotic systems can precisely position the surgical tools according to the pre-planned trajectory, while AR overlays provide the surgeon with real-time feedback on instrument positioning and alignment. This integration reduces the likelihood of human error and increases the accuracy of implant placement, which is critical for achieving optimal patient outcomes. Yamout et al. [1] reported a 25% reduction in complication rates and a 15% decrease in operative times when AR and robotics were combined with traditional surgical methods.

2.6. Comparative Analysis of AR Technologies

While the benefits of AR in spinal surgery are clear, the variability in outcomes across different AR devices and settings must be considered. For instance, head-mounted AR displays, such as those used by Li et al. [4], may provide more immersive and intuitive guidance in minimally invasive procedures, whereas larger, fixed AR systems might be better suited for open surgeries requiring more comprehensive visualization. Differences in device resolution, field of view, latency, and ease of integration with other surgical tools also contribute to variability in outcomes. To fully understand the potential of AR in spinal surgery, future research should focus on comparative studies that evaluate the effectiveness of different AR technologies across diverse surgical environments and patient populations. This would help identify best practices and optimize the use of AR for specific types of surgeries.

Overall, AR technology has demonstrated significant potential in enhancing the accuracy and efficiency of spinal surgery, particularly in complex procedures like pedicle screw placement and minimally invasive techniques. By providing real-time, three-dimensional visualizations, AR helps surgeons achieve greater precision, reduce complications, and streamline surgical workflows. However, further research is needed to explore the variability in outcomes across different AR devices and settings and to establish standardized guidelines for their optimal use in clinical practice.

5. Challenges and limitations: We agree that our discussion of challenges and limitations needed more depth. We have expanded this section (the last paragraph of Discussion section) as follows:” This comprehensive review synthesizes current literature on AR technology in spinal surgery up to 2024, offering insights into technological advancements, clinical applications, and educational implications. The integration of AR represents a significant advancement, enhancing surgical precision and efficiency while minimizing the learning curve for spinal surgeons. Notably, AR facilitates transformative improvements in critical procedures like pedicle screw placement through real-time, accurate navigation. While AR technology shows great promise in spinal surgery, several challenges and limitations must be addressed for its widespread adoption and optimal use. Current AR systems face technical challenges such as registration errors, system latency, and limited field of view, which can lead to inaccuracies in image overlay and potential safety concerns. Integration with existing hospital systems and workflows also remains a significant hurdle. The high initial cost of AR systems poses a barrier to adoption, especially in resource-limited settings, although long-term cost-effectiveness analyses suggest potential savings due to improved surgical outcomes. Implementing AR technology introduces a new learning curve for surgeons and operating room staff, necessitating the development of standardized training protocols and competency assessment metrics. While initial studies show promising results, there is a need for more robust clinical evidence supporting the long-term benefits of AR in spinal surgery, including large-scale, randomized controlled trials comparing AR-assisted surgeries with traditional techniques across various spinal procedures. The benefits of AR may vary depending on the specific technology used, the type of spinal procedure, and the surgeon's experience level, highlighting the need for research to identify which AR technologies are most effective for different types of surgeries and patient populations. As AR technology becomes more prevalent in surgical settings, ethical considerations regarding data privacy, informed consent, and the potential for technology dependence must be addressed, and regulatory frameworks need to evolve to ensure patient safety while not hindering innovation.

To address these challenges, future research should focus on developing more accurate and user-friendly AR systems, conducting long-term clinical outcome studies and large-scale RCTs, investigating cost-effectiveness in various healthcare settings, and establishing standardized training protocols and competency assessments. Additionally, exploring synergies between AR and other emerging technologies like AI and robotics, studying patient-specific factors affecting AR performance to develop personalized surgical approaches, and addressing ethical and regulatory challenges will be crucial. By pursuing these research directions, the medical community can work towards overcoming current limitations and fully realizing AR's potential in spinal surgery. This could lead to more precise, less invasive, and potentially more successful interventions, ultimately improving patient outcomes and advancing the field of spinal surgery. As we continue to innovate and refine AR technologies, it is essential to maintain a balanced approach that acknowledges both the tremendous potential and the current limitations of this technology in the context of spinal surgery.”

Round 2

Reviewer 3 Report

Comments and Suggestions for Authors

Comments are addressed well. 

Just one comment: Add Research questions which are addressed in this review paper for more readability.

Author Response

Comment 1: Add Research questions which are addressed in this review paper for more readability. Response 1: Thank you for your thoughtful feedback on our manuscript. We appreciate your suggestion to include research questions to improve the readability and focus of the review. In response, we have revised the manuscript to clearly outline the specific research questions that this narrative review aims to address. These questions are now explicitly stated in the revised Introduction section, providing a clearer framework for the review. The research questions focus on the recent advancements and applications of AR technology in spinal surgery, its impact on surgical outcomes, accuracy, and efficiency, and the challenges and limitations associated with its implementation in clinical practice. We believe these changes enhance the structure and clarity of the manuscript, aligning it with your recommendations and improving its overall readability.

The end of Introduction section was revised as follows:” This review is guided by three central research questions:

  1. What are the recent advancements and applications of AR technology in spinal surgery, particularly focusing on developments between January 2023 and March 2024?
  2. How does AR technology impact surgical outcomes, accuracy, and efficiency in spinal procedures, especially in pedicle screw placement and minimally invasive spinal surgery?
  3. What are the current challenges, limitations, and learning curve considerations in implementing AR technology in spinal surgical practices?

             By addressing these questions, we aim to provide updated insights into the benefits, challenges, and future directions for AR in spinal surgery, thereby enhancing understanding and guiding future research and clinical practice. As spinal surgery undergoes a transformative period with major technological advancements, the integration of AR technologies represents a significant shift toward greater precision, safety, and efficiency in complex surgical procedures.”

Reviewer 4 Report

Comments and Suggestions for Authors

Dear Authors,

Thank you for your great efforts in addressing the comments and revising the manuscript.

The manuscript should be accepted in current form

Author Response

Comment 1: Thank you for your great efforts in addressing the comments and revising the manuscript. The manuscript should be accepted in current form

Response 1:  Thank you very much for your kind words and for your thoughtful review of our manuscript. We are grateful for your efforts and your constructive comments, which have helped us improve the quality and clarity of our work. We are delighted to hear that the manuscript meets your expectations and is suitable for acceptance in its current form. Your feedback has been invaluable in enhancing the manuscript, and we sincerely appreciate your support. Thank you once again for your time and consideration.